

# Simulation of electroporation threshold based on the evolution of transmembrane potential and pore density

Yu Zhang[1], Zhijun Luo[2] and Fei Guo[2]

[1] Department of Gynecology, Chongqing Hospital of Traditional Chinese Medicine, Chongqing, China
[2] College of Automation, Chongqing University of Post and Telecommunications, Chongqing, China

## ABSTRACT

To study the electric field threshold of electroporation of real cell membrane structures under the action of the pulsed electric field, in this article, a finite element model of the real cell containing endoplasmic reticulum and nucleus was constructed in real cell staining images by cluster segmentation and edge extraction techniques. The electroporation equation was introduced into the real cell model to calculate the threshold value of different membrane structures for electroporation under a pulsed electric field. The results showed that the transmembrane potentials of the cell membrane, endoplasmic reticulum membrane, and nuclear membrane reached the electroporation thresholds at 1.2, 2.6, and 2.9 kV/cm, while the pore density thresholds were $1.7 \times 10^{14}/m^2$, $3.2 \times 10^{14}/m^2$, and $3.5 \times 10^{14}/m^2$, respectively. Under a single pulse with a pulse width of 100 µs and rise and fall times of 10 µs, the pore density reaches the electroporation threshold at 1.7, 3.2 and 3.5 kV/cm, respectively.

## INTRODUCTION

Electroporation (EP) is a biophysical phenomenon in which an applied high-intensity pulsed electric field increases the permeability of biological cell membranes, thereby facilitating the uptake of biomolecules into the cell (*Kotnik et al., 2019*). Electroporation can be used to facilitate also ingress of ions and other small molecules, *e.g.*, cytostatics. Currently, technologies based on the EP effect have been widely used in the fields of gene transfection (*Kotnik et al., 2019*), cell fusion (*Dermol-Černe, Pirc & Miklavčič, 2020*), drug introduction (*Sweeney et al., 2016*), and environmental sterilization (*Hasija et al., 2021*).

Despite widespread use, optimization of pulse parameters in electroporation remains a critical research focus. Reversible electroporation (RE) is standardized in Europe for electrochemotherapy, which was called European Standard Operating Procedures of Electrochemotherapy (ESOPE) (*Marty et al., 2006*; *Gehl et al., 2018*), while irreversible electroporation (IRE) is widely used in the USA, with FDA approval (*Bendix, 2022*). Clinical studies show IRE's efficacy and safety in treating various tumors. However, continuous refinement of pulse parameters (*e.g.*, voltage, pulse duration, and frequency) is still needed to enhance clinical outcomes and adapt to diverse therapeutic needs. The main

Corresponding authors
Yu Zhang, yuyu_xiaxia@163.com
Fei Guo, guofei@cqupt.edu.cn

reason is that the microscopic mechanism of electroporation is still unclear (*Dermol-Černe, Pirc & Miklavčič, 2020*). Therefore, the prediction of the EP threshold of individual cells remains an urgent challenge.

The occurrence of cellular EP is closely related to the parameters of the applied pulse, and it is usually considered that the occurrence of EP is a threshold effect, and the threshold is generally determined by the field strength of the applied pulsed electric field (*Polak et al., 2015*). Recent studies have found that the electric field threshold is not only related to the electric field strength but also depends on the pulse duration (*Wyss et al., 2024*). Therefore, researchers have mostly used the value of applied pulse field strength and the pulse duration to characterize the occurrence and development of electroporation in experimental studies (*Ezzeddine, Asirvatham & Nguyen, 2024*).

Researchers have attempted to directly observe the formation, development, and disappearance of electroporation from an experimental point of view to obtain the electroporation threshold of cells. Due to the inherent shortcomings in the spatial and temporal resolution and sensitivity of the research methods and equipment, as well as the complex dynamics of biological cells, no convincing and reliable conclusions have yet been obtained (*Sozer et al., 2020*). Scholars in this field have turned to theoretical simulation and other means to reveal the microscopic mechanism of EP, based on which research methods such as molecular dynamics simulation (*Rems et al., 2020*), equivalent circuit model (*Brosseau & Sabri, 2021*), mesh transport method (*Mi et al., 2019*), and finite-element calculations (*Chiapperino et al., 2020*) have been generated.

Among them, the finite element calculations, which employs finite element analysis to solve Maxwell's equations for dielectric materials under pulsed electric fields, is widely adopted in electroporation research. This approach enables adaptive spatial and temporal resolution for multi-scale simulations and facilitates efficient coupling of multi-physics phenomena (*Jiang, Davalos & Bischof, 2015*). Our group has carried out a lot of simulation work on the mechanism of electroporation (*Zhang et al., 2018*; *Guo et al., 2020, 2019*), however, most of the previous studies are based on spherical cells and less on the mechanism of electroporation in real cells. The studies involving real cells did not include the threshold study of electroporation in intracellular structures such as endoplasmic reticulum membranes.

Therefore, based on previous studies, this article investigates the EP threshold of cells under the action of typical microsecond pulses through the spatial and temporal distributions of TMP and microporosity densities based on real cells containing intracellular structures.

## SIMULATION MODEL

Figure 1 shows a fluorescence microscopy image of DC-3F cells (Chinese hamster fibroblasts), in which the nucleus was stained blue and the endoplasmic reticulum was stained green (the DC-3F cells were obtained from the Stem Cell Bank, Chinese Academy of Sciences, Beijing, China) The nucleus dye was DAPI and the endoplasmic reticulum dye was ER-Tracker Green. The stained image was subjected to cluster segmentation and edge

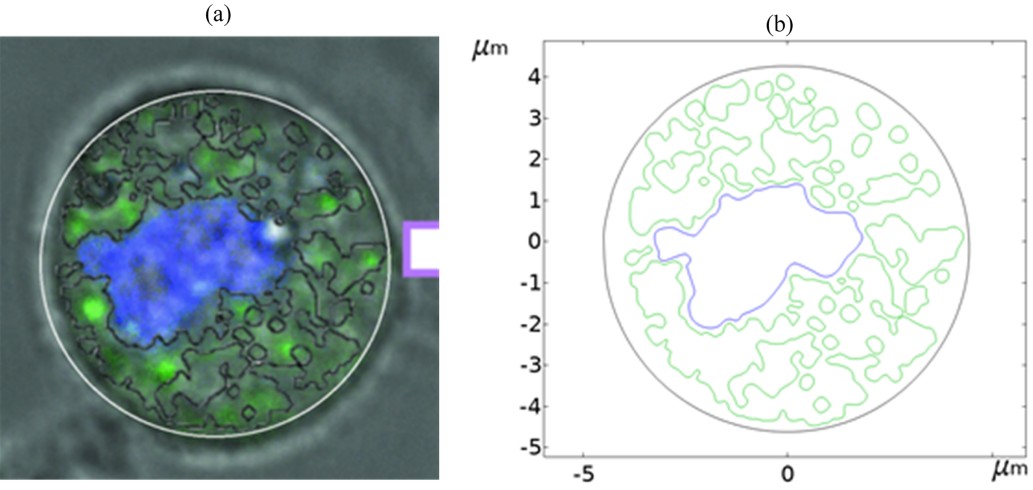

**Figure 1 Diagram of realistic cell staining and boundary extraction.**

extraction to obtain its boundary bitmap (*Denzi et al., 2016*), which was converted to a vectorial image and then imported into the COMSOL Multiphysics 5.6 software (COMSOL Inc., Sweden) to construct a two-dimensional finite element dielectric model of real cells (Fig. 2). The model contains extracellular fluid, cell membrane, cytoplasm, nuclear membrane, nucleoplasm, endoplasmic reticulum membrane, and endoplasmic reticulum matrix, and the geometric dimensions and electromagnetic parameters of each part are detailed in Table 1.

As shown in Fig. 2, the left and right sides of the rectangle represent the flat electrodes to which the electric field is applied during the simulation calculation. A pulsed power supply is applied to the left side and the right side is set to zero potential, thus creating a pulsed electric field between the flat electrodes. The top and bottom sides of the rectangle are set to be electrically insulated. The entire calculation will be kept current-conserving and the upper and lower pole plates are set to be electrically insulated. The initial value of the potential at each point on the cell is zero. The model uses current modules and coefficient partial differential equations. The current module is used to calculate the potential distribution at various points in the model and the coefficient partial differential equation is used to calculate the variation of pore density. The mesh is optimally dissected and contains 107,872 domain cells and 4,757 boundary cells.

To study the spatial and temporal distribution of the TMP and the density of pores, a point and a segment of arc length were selected as observation points on different structural membranes in the cell, as shown in Fig. 3. The transmembrane potential is calculated at the selected point and on the continuous arc length, respectively. The membrane structures in this cell model contain cell membrane, nuclear membrane, and endoplasmic reticulum membrane. The cell membrane and nuclear membrane contain one closed arc length, while the endoplasmic reticulum membrane contains 36 closed arc
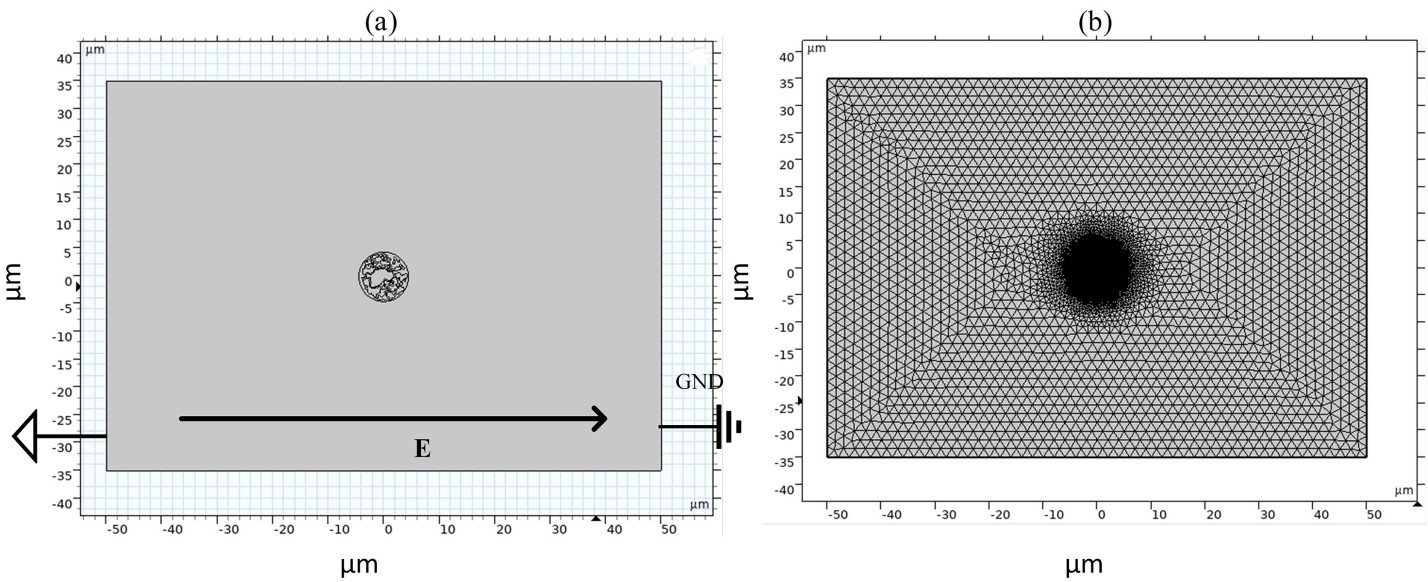

**Figure 2 Diagram of simulation model and grid segmentation.** (A) Geometric structure. (B) Mesh sectioning.

**Table 1 Cell parameters used in calculation.**

| Parameters | Cellular component | Value | Reference |
|---|---|---|---|
| Geometric parameter (μm) | Extracellular fluid | 100 × 70 | *Guo et al. (2019)* |
| | Cell membrane thickness | 0.01 | *Guo et al. (2020)* |
| | Nuclear membrane thickness | 0.04 | *Luo, Guo & Gong (2025)* |
| | Endoplasmic reticulum membrane thickness | 0.01 | *Guo et al. (2020)* |
| Conductivity (S/m) | Extracellular fluid | 0.55 | *Guo et al. (2019)* |
| | Cell membrane | $1.1 \times 10^{-7}$ | *Guo et al. (2019)* |
| | Cytoplasm | 0.55 | *Guo et al. (2019)* |
| | Nuclear membrane | $1.1 \times 10^{-5}$ | *Guo et al. (2019)* |
| | Nuclear matrix | 0.55 | *Guo et al. (2019)* |
| | Endoplasmic reticulum membrane | $1.1 \times 10^{-7}$ | *Guo et al. (2019)* |
| | Endoplasmic reticulum matrix | 0.55 | *Guo et al. (2019)* |
| Relative permittivity | Extracellular fluid | 67.00 | *Guo et al. (2019)* |
| | Cell membrane | 5 | *Guo et al. (2019)* |
| | Cytoplasm | 67.00 | *Guo et al. (2019)* |
| | Nuclear membrane | 5 | *Guo et al. (2019)* |
| | Nuclear matrix | 67.00 | *Guo et al. (2019)* |
| | Endoplasmic reticulum membrane | 5 | *Guo et al. (2019)* |
| | Endoplasmic reticulum matrix | 67 | *Guo et al. (2019)* |
| | Electroporation parameters ($\alpha$) | $1.0 \times 10^{9}$ (1/(m² × s)) | *Guo et al. (2020)* |
| | Initial pore density when $V_m = 0 (N_0)$ | $1.5 \times 10^{9}$ (1/m²) | *Guo et al. (2020)* |
| | Characteristic voltage ($V_{ep}$) | 0.258 (V) | *Guo et al. (2020)* |
| | Electroporation constant ($q$) | 2.46 | *Guo et al. (2020)* |
| | Pore radius ($r_p$) | 0.76 (nm) | *Guo et al. (2020)* |

| Table 1 (continued) | | | |
| --- | --- | --- | --- |
| **Parameters** | **Cellular component** | **Value** | **Reference** |
| Electroporation parameters | Porous energy barrier ($w_0$) | 2.65 | *Guo et al. (2020)* |
| | Porous conductivity ($\sigma_p$) | 1.3 (S/m) | *Guo et al. (2020)* |
| | Porous inlet length ($n$) | 0.15 | *Guo et al. (2020)* |
| | Temperature ($T$) | 295 (K) | *Guo et al. (2020)* |
| | Gas constant ($R$) | 8.314 (J/K/mol) | *Guo et al. (2020)* |
| | Faraday's constant ($F$) | $9.65 \times 10^4$ (C/mol) | *Guo et al. (2020)* |

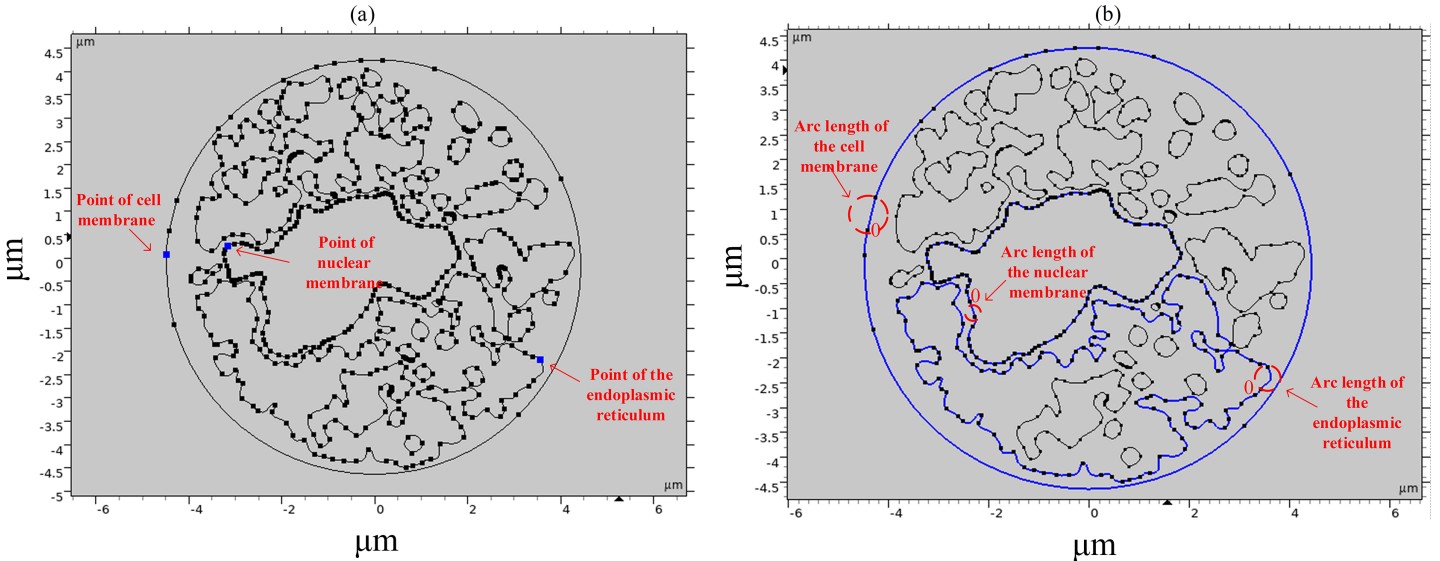

**Figure 3 Point and arcs selected in the simulation model.**

lengths, and the largest closed arc length was selected to observe the spatial distribution of TMP and microporosity.

Typical values of the most widely used cellular parameters in the literature were chosen as the parameters for simulation calculations (*Zhang et al., 2018*; *Guo et al., 2019*, *2020*; *Denzi et al., 2016*), which are remarkably representative, and for the reader's convenience, each parameter is listed in Table 1.

EP including the formation and development of pores, is generally described by the Smoluchowski partial differential equation, which is explained in detail in the literature (*Neu & Krassowska, 1999*) and is briefly described in this article as follows. The conductivity $\sigma_m(t)$ and the microporosity density $N(t)$ of the different membrane structures after the EP effect occurs are *Krassowska & Filev (2007)*:

$$\sigma_m(t) = \sigma_{m0} + N(t)\pi r_p^2 \sigma_p K \tag{1}$$

$$\sigma_p = \frac{2\sigma_e \sigma_i}{\sigma_e + \sigma_i} \tag{2}$$

$$\frac{dN(t)}{dt} = \alpha e^{\left(\frac{V_m(t)}{V_{ep}}\right)^2}\left(1 - \frac{N(t)}{N_0}e^{-q\left(\frac{V_m(t)}{V_{ep}}\right)^2}\right). \qquad (3)$$

The electric field distribution incorporating the cell membrane EP effect was calculated using the following equation:

$$-\nabla \cdot \frac{\partial}{\partial t}(\varepsilon \nabla \phi) - \nabla \cdot \sigma \nabla \phi = 0 \qquad (4)$$

$$E = -\nabla \varphi. \qquad (5)$$

The transmembrane potential is the difference between the outer membrane potential $\phi_o(t)$ and the inner membrane potential $\phi_i(t)$ of the cell or nuclear membrane:

$$\Delta \varphi = \varphi_o(t) - \varphi_i(t). \qquad (6)$$

The temporal and spatial distributions of TMP and pore densities of different membrane structures can be obtained by associating Eqs. (1)–(5) to include the EP effect.

The model physical field is selected from the current module and the coefficient partial differential equation module. The former calculates the electric field distribution, which is solved over the entire simulation region. The latter is used to solve for the pore density $N$ in Eq. (2), and the calculation of $N$ is performed in the region of the cell membrane, the nuclear membrane, and the endoplasmic reticulum membrane.

## RESULTS

To study the electric field thresholds of different membrane structures of single cells under a microsecond pulsed electric field, typical pulsed electric field parameters were chosen in this article, rectangular pulses with a pulse width of 100 us and rising and falling edges of 10 μs. The average pore radius under EP threshold conditions was estimated as ~1–2 nm based on the Smoluchowski equation (Eq. (3)), consistent with previous molecular dynamics simulations (*Retelj, Pucihar & Miklavcic, 2013*).

In this article, the following ideas are used to obtain the electric field thresholds of different membrane structures during the action of pulsed electric fields. A transmembrane potential exceeding 1 V or a pore density of $10^{14}/m^2$ is widely accepted as the threshold for electroporation occurrence. In modeling calculations, we usually consider the initial pore density of the cell membrane to be $1.5 \times 10^9$ (*Mao et al., 2018*). The electric field was applied in steps of 0.1 kV/cm starting from zero, and the spatial and temporal distributions of the transmembrane potential and porous density were observed at all moments over the membrane arc length to obtain the electric field threshold corresponding to the transmembrane potential or porous density.

When the applied electric field was increased to 1.2 kV/cm, regions with a transmembrane potential greater than 1 V began to appear on the cell membrane, and the higher the transmembrane potential the higher the pore density, as shown in Figs. 4A, 4B. By analyzing the time evolution patterns of transmembrane potential and pore density at three points on different membrane structures, it was found that the cell membrane transmembrane potential was similar to the pattern of change of the applied pulse, as shown in Figs. 4C, 4D. The transmembrane potential reached a flat peak value of about

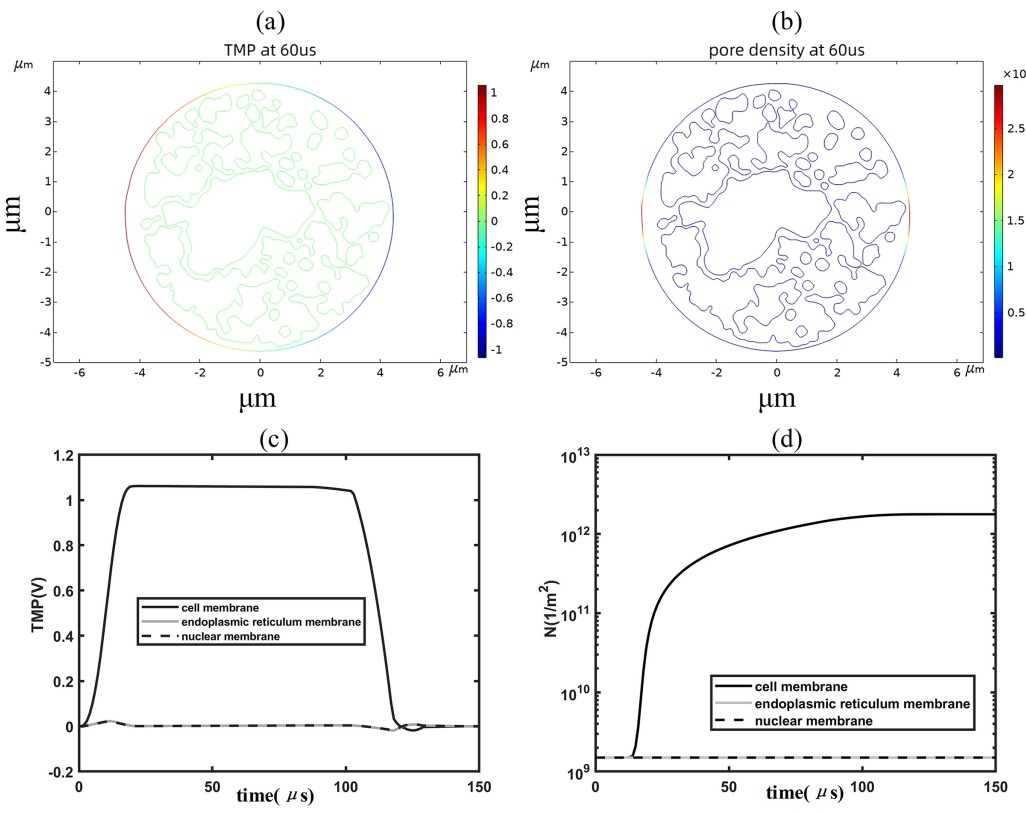

**Figure 4 Changes in electroporation indicators.** (A) and (B) 2D distribution of TMP and N (C) and (D) Time course of TMP (C) and N (D) of points on cell membranes.

1.05 V at 20 μs and started to decrease at about 100 μs. The pore density of the cell membrane increases with time and reaches about $2 \times 10^{12} \mathrm{m}^{-2}$ at 100 μs. The temporal evolution of the transmembrane potential and pore density of the cell membrane match well with that of previous studies, which is due to the relatively close spherical shape of the cell membrane in the present simulation model (*Brosseau & Sabri, 2021*; *Mi et al., 2019*; *Guo et al., 2019*). In addition, the transmembrane potential and pore density of the nuclear and endoplasmic reticulum membranes did not change much under the effect of 1.2 kV/ cm field strength.

To further analyze the spatial distribution of transmembrane potential and pore density in different membrane structures, the spatial distribution of transmembrane potential and pore density along the arc length under 1.2 kV/cm field strength is also given in this article, as shown in Fig. 5. The distribution of TMP along the arc length of the cell membrane was following the cosine law (*Brosseau & Sabri, 2021*):

$$V_m(\theta) = \frac{3}{2} E R cos(\theta). \tag{7}$$

The density distribution of pores also showed spatial symmetry, which was in good agreement with the results of the previous studies on spherical cells (*Mi et al., 2019*;

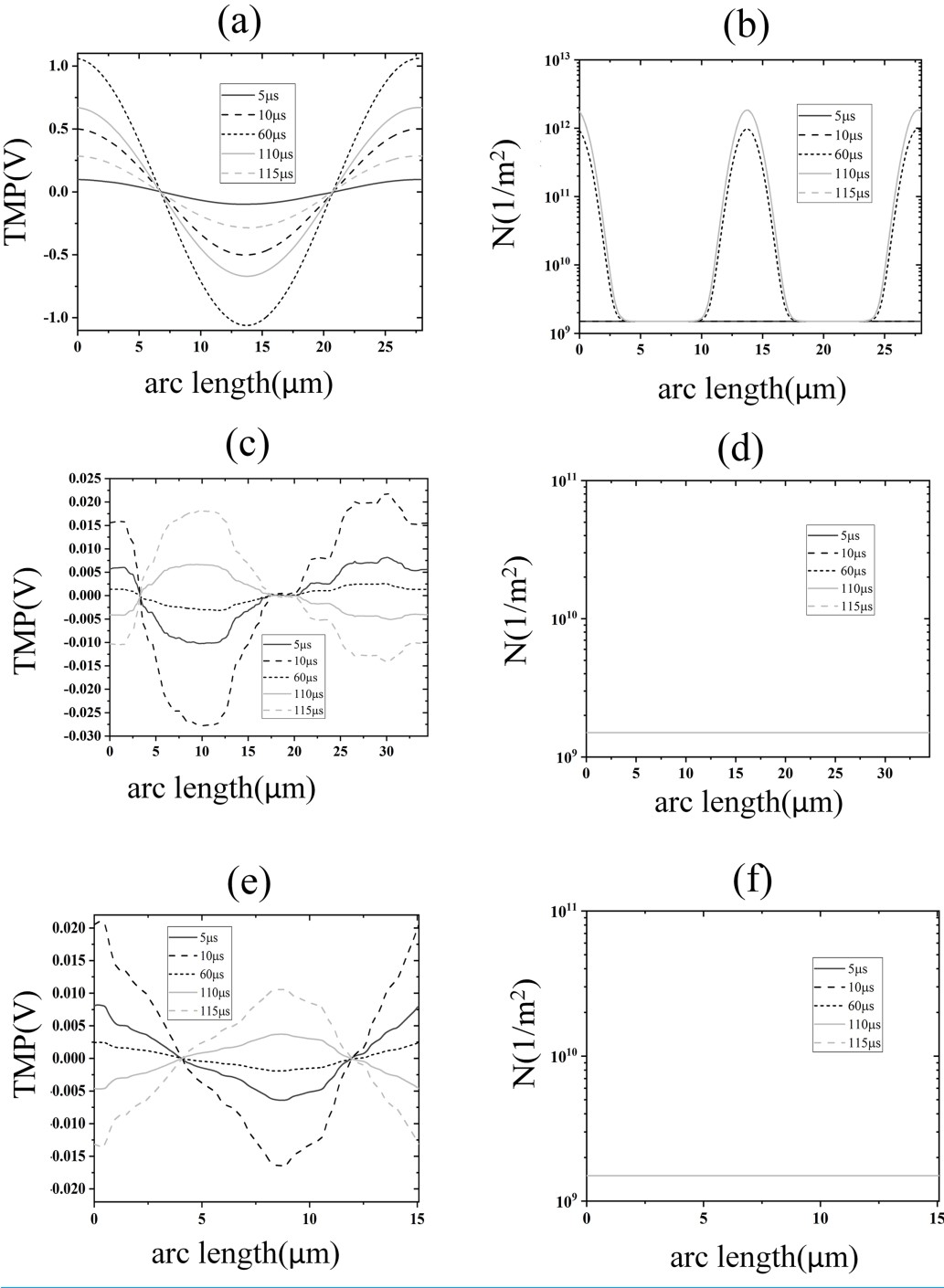

**Figure 5 Electroporation indicators for different membrane structures.** (A) and (B) Arc-length distribution of TMP and N along the plasma membrane. (C) and (D) Arc-length distribution of TMP and N along the endoplasmic reticulum membrane. (E) and (F) Arc-length distribution of TMP and N along the nuclear membrane.

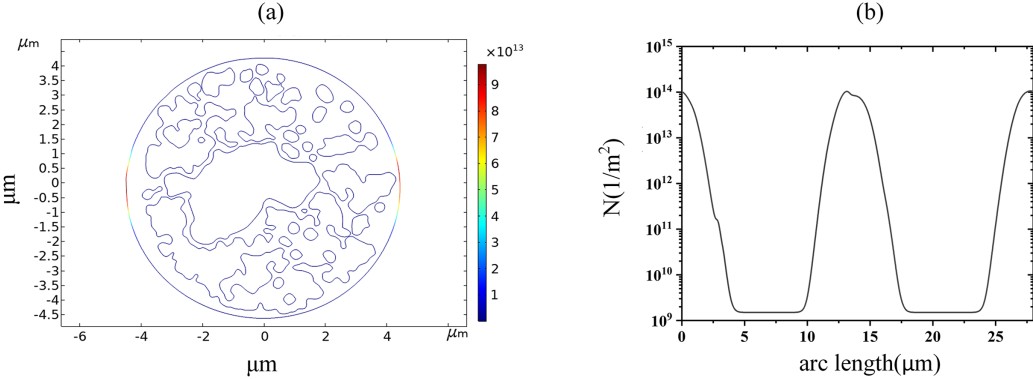

**Figure 6 Variation in pore density along the arc length distribution.** (A) 1.7 kV/cm-60us moment pore density in 2D. (B) pore density along the arc length of the plasma membrane.

*Zhang et al., 2018*). This behavior arises because the near-spherical geometry of the cell membrane allows the transmembrane potential to follow the classical cosine distribution law under homogeneous electric fields, as predicted by analytical models for spherical cells (*Mi et al., 2019*; *Zhang et al., 2018*). At the electroporation threshold (1.2 kV/cm), pore density remains low, leaving membrane conductivity largely unaltered and preserving the TMP symmetry. In contrast, the nuclear and endoplasmic reticulum membranes exhibit irregular geometries, leading to localized electric field enhancements and asymmetric TMP distributions that deviate from the cosine law (*Mi et al., 2019*). Notably, although pore densities in these intracellular membranes remain subthreshold ($<10^{14}/m^2$) at 1.2 kV/cm, the transient TMP spikes during pulse edges suggest that high-frequency components of the pulse preferentially polarize intracellular membranes—a phenomenon consistent with studies demonstrating nanosecond pulses' selective effects on organelle electroporation (*Esser et al., 2010*). This geometric dependency underscores that EP thresholds and TMP dynamics are intrinsically linked to membrane topology, as emphasized in multi-scale simulations of non-spherical cells (*Denzi et al., 2016*). The simulation results of this part show that under the effect of 1.2 kV/cm field strength, the cell membrane began to undergo EP, but at this time the spatial distribution of the TMP of the cell membrane did not change significantly, and the pore density did not reach the EP threshold mentioned in the literature. The TMP and pore density of nuclear and endoplasmic reticulum membranes were lower, and neither of them experienced EP.

With the increase of the applied pulsed electric field strength, the pore density of the cell membrane reached the EP threshold of $10^{14}m^{-2}$ at 1.7 kV/cm, which can be seen by the spatial and along-arc length distribution of the pore density at 60 μs, as shown in Fig. 6. At this time, the pore density was no longer symmetrically distributed along the arc length, indicating that the cell membrane undergoes an obvious EP effect. A large number of ions in the cellular solution begin to enter the intracellular space.

As described above, the EP threshold electric fields of endoplasmic reticulum membranes were obtained as 2.6 and 3.2 kV/cm, respectively, with the former indicating

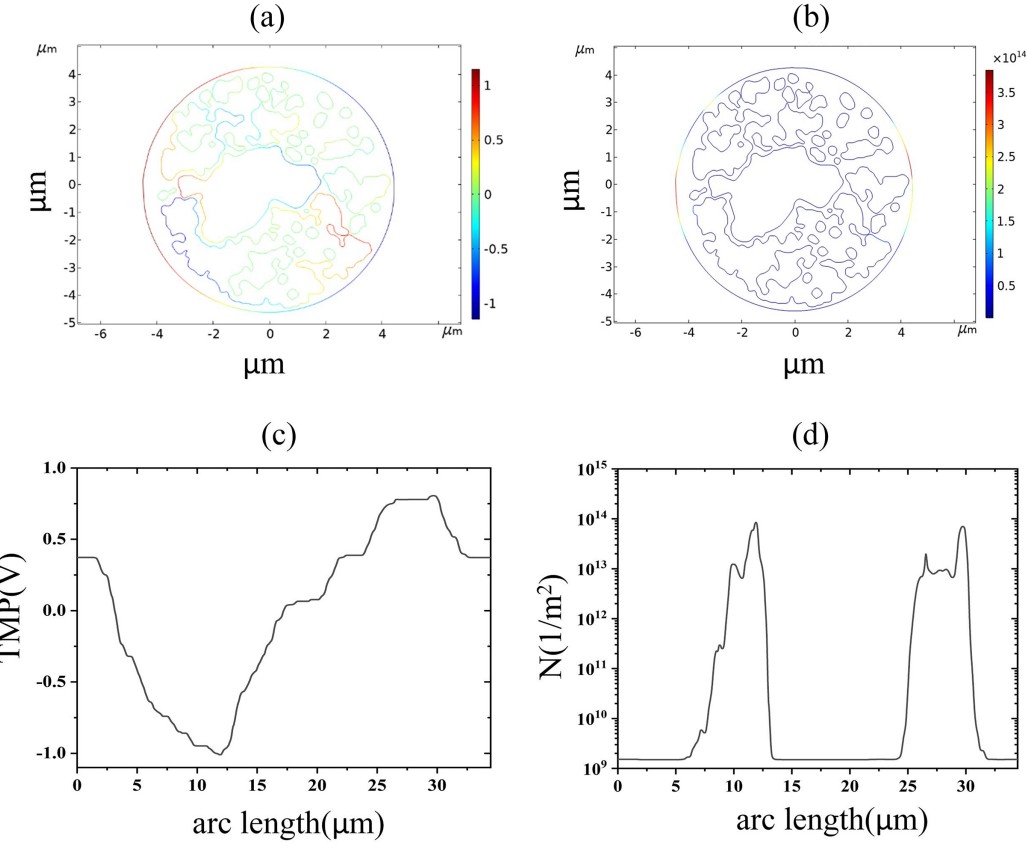

**Figure 7 Endoplasmic reticulum membrane.** (A) 2.6 kV/cm-60µs moment TMP in 2D. (B) 3.2 kV/cm-60µs moment pore density in 2D. (C) Distribution of TMP along the arc length at the moment of 2.6 kV/cm-60µs. (D) Distribution of pore density along the arc length at the moment of 3.2 kV/cm-60µs

the beginning of the appearance of regions with TMP greater than 1 V in the endoplasmic reticulum membrane and the latter indicating the beginning of the appearance of regions with pore densities greater than $10^{14}\text{m}^{-2}$ in the endoplasmic reticulum membrane. When 2.6 kV/cm pulsed electric field action, the cell membrane and endoplasmic reticulum membrane both appear transmembrane potential greater than 1 V region, and the distribution of transmembrane potential along the endoplasmic reticulum membrane arc length is the same as 1.2 kV/cm, indicating that at this time, the spatial distribution of the transmembrane potential has not been significantly changed, as shown in Fig 7. When 3.2 kV/cm pulsed electric field action, the density of pores in the endoplasmic reticulum membrane reached $10^{14}\text{m}^{-2}$ and coincided with the distribution of the TMP, and both reached the maximum absolute value in the same area, which again indicated that the occurrence of electroporation was closely related to the membrane structure.

The nuclear membrane reached the thresholds of TMP and pore density at 2.9 and 3.5 kV/cm, respectively, as shown in Fig. 8. The distribution of TMP and pore density in nuclear membranes is strongly correlated, with larger TMP corresponding to higher pore densities. The distribution of TMP and pore density is consistent with the above

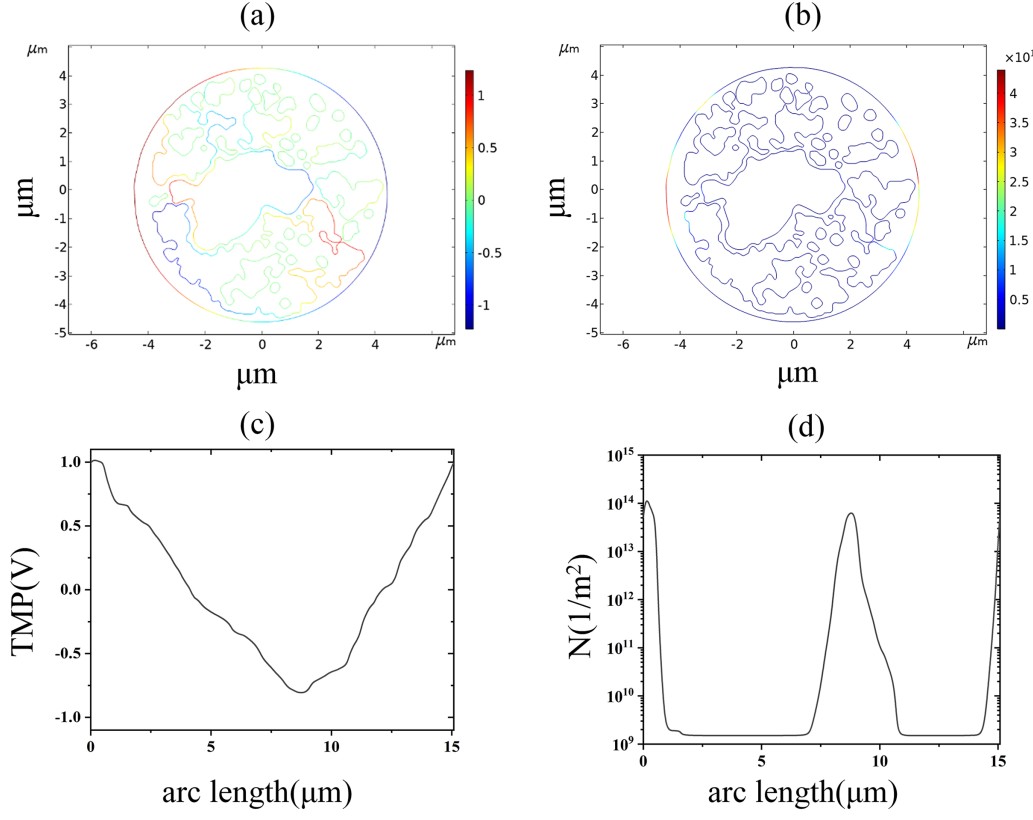

**Figure 8 Nuclear membrane.** (A) 2.9 kV/cm-60μs moment TMP in 2D. (B) 3.5 kV/cm-60μs moment pore density in 2D. (C) Distribution of TMP along the arc length at the moment of 2.9 kV/cm-60μs. (D) Distribution of pore density along the arc length at the moment of 3.5 kV/cm-60μs.

distribution patterns in cell membranes and endoplasmic reticulum membranes, both of which are closely related to the membrane structure. In addition, the field strength required for nuclear membrane EP was the highest, and both cellular and endoplasmic reticulum membranes were already electroporated when nuclear membrane EP occurred.

# DISCUSSION

Although EP-based technologies have been widely used, there is still a large blindness in the selection of pulse parameters in use, which seriously restricts the promotion and application of this technology, the reason for this is that the micro-mechanism of EP is still unclear (*Kotnik et al., 2019*; *Brosseau & Sabri, 2021*; *Jiang, Davalos & Bischof, 2015*). From the time scale, the generation, development, and disappearance of EP is a biophysical process involving nanoseconds, microseconds, seconds, and even minutes and hours across the time scale. On a spatial scale, electroporation involves cross-space size distributions on the nanometer, micrometer, and even centimeter scale. From the perspective of the underlying biophysical mechanisms, EP research involves multi-physical field interdisciplinary fields such as electric field, temperature field, biomechanics,

biochemistry, and physiology. Therefore, the study of the EP mechanism belongs to the interdisciplinary problem of multi-physical field coupling at multi-temporal and spatial scales, which is also the biggest problem that has been troubling the field for decades.

The study of EP at the single-cell level is still a hotspot. It is difficult to effectively refine the threshold parameter for inducing cellular EP because the occurrence of EP in a single cell is closely related to the cellular structure, the electric field parameters, the surrounding solution parameters, and the parameters of the applied pulses, which involve numerous variables (*Jiang, Davalos & Bischof, 2015*). From an experimental point of view, the field strength of the applied electric field is a key parameter and is also considered to be the most important parameter in determining the occurrence and development of EP (*Polak et al., 2015*). Therefore, the applied electric field strength is commonly used in clinical applications to measure the degree of occurrence and development of EP.

From the perspective of cellular response, the pulsed electric field produces a transmembrane potential and a large number of pores in the membrane structure of the cell, the former is often regarded as an indication for the generation of EP, and the latter is a prerequisite for the uptake of molecules into the cell. It is usually believed that the cell produces an EP effect when the transmembrane potential reaches 1 V (*Jiang, Davalos & Bischof, 2015*). In contrast, some scholars have argued from the perspective of ion transport that cells undergo EP when the pore density of the membrane structure on the cell reaches $10^{14}\mathrm{m}^{-2}$ (*Retelj, Pucihar & Miklavcic, 2013*). Therefore, in this article, transmembrane potential and pore density were chosen as the parameters indicative of the occurrence of the EP effect in the supracellular membrane structure.

From the perspective of transmembrane potential, EP occurred at 1.2, 2.6, and 2.9 kV/cm for cell membrane, endoplasmic reticulum membrane, and nuclear membrane, respectively. From the perspective of pore density, EP occurred at 1.7, 3.2, and 3.5 kV/cm for the cell membrane, endoplasmic reticulum membrane, and nuclear membrane, respectively. Therefore, a pore density of $10^{14}\mathrm{m}^{-2}$ is a more stringent index to characterize EP generation. In addition, the simulation results in this article show that the transmembrane potential is closely related to the pore density distribution, and both are closely related to the geometrical features of the membrane structure, which show strong local characteristics. The membrane structure electroporation threshold is inversely proportional to the geometric size, the larger the membrane structure size, the lower its electroporation threshold, which is also consistent with previous studies (*Mi et al., 2019*; *Chiapperino et al., 2020*).

While *Milestone et al. (2023)* focused on multi-organelle interactions and proximity effects in clustered cells (*Baker et al., 2024*), our study uniquely quantifies EP thresholds for individual subcellular structures in real cell geometries derived from fluorescence imaging. Additionally, we explicitly couple pore density dynamics with TMP evolution, enabling direct comparison of EP criteria, which was not addressed in their framework. In this article, the EP thresholds of different membrane structures under 100 ns pulse width and 10 ns rising and falling edge pulses are also given in Table 2. Consistent with the simulation

**Table 2 Field threshold of electroporation.** Simulation results of electric field threshold of electroporation.

| 100 μs (100 ns) | The field strength required for the transmembrane potential to reach 1V (kV/cm) | The field strength required for the microporous density to reach $10^{14}$ (kV/cm) |
|---|---|---|
| Cell membrane | 1.2 (2.4) | 1.7 (3.2) |
| Endoplasmic reticulum membrane | 2.6 (4.8) | 3.2 (6) |
| Nuclear membrane | 2.9 (3.8) | 3.5 (4.9) |

results at a pulse width of 100 μs, the pore density needs higher field strength to reach the threshold, but the field strength at this time is higher than that of the results at 100 μs. The nuclear membrane is electroporated before the endoplasmic reticulum membrane under 100 ns pulses. As the applied electric field pulse width is further reduced, the nuclear membrane may be the first to be perforated, coinciding with the effect of electro-processing within nanosecond pulses (*Esser et al., 2010*).

In this article, real cells containing intracellular structures are used to obtain the electric field thresholds for EP of different membrane structures based on the transmembrane potential and pore density, which is obvious progress compared with the previous simulation studies of spherical cells and real cells containing only cell membranes, however, the electromagnetic parameters of each cell component and the parameters of EP in this article are from the classical literature rather than the real measurements, and there will be an error in predicting the EP of cells. In conclusion, the method provided in this article can guide single-cell EP threshold prediction.

## CONCLUSION

In this article, we investigated the electric field threshold of EP in real cells containing several types of membrane structures. The results show that there is a strong correlation between the spatial and temporal distribution of the transmembrane potential and the density of pores, both of which are closely related to the geometrical features of the membrane structure. Based on the transmembrane potential threshold, EP occurred at 1.2, 2.6, and 2.9 kV/cm for the cell membrane, endoplasmic reticulum membrane, and nuclear membrane, respectively. Based on the pore density threshold, the cell membrane, endoplasmic reticulum membrane, and nuclear membrane underwent EP at 1.7, 3.2, and 3.5 kV/cm, respectively.

### Funding
This work was supported by the National Natural Science Foundation of China (No. 52377223) and the Chongqing Science and Technology Commission Incentive and Guidance Project (cstc2021jxjl130008). The funders had no role in study design, data collection and analysis, decision to publish, or preparation of the manuscript.

## Grant Disclosures

The following grant information was disclosed by the authors:
National Natural Science Foundation of China: 52377223.
Chongqing Science and Technology Commission Incentive: cstc2021jxjl130008.

## Competing Interests

The authors declare that they have no competing interests.

## Author Contributions

- Yu Zhang conceived and designed the experiments, performed the experiments, analyzed the data, authored or reviewed drafts of the article, and approved the final draft.
- Zhijun Luo conceived and designed the experiments, performed the experiments, analyzed the data, prepared figures and/or tables, authored or reviewed drafts of the article, and approved the final draft.
- Fei Guo conceived and designed the experiments, performed the experiments, authored or reviewed drafts of the article, and approved the final draft.

## Data Availability

The raw data is available in the Supplemental File.

## Supplemental Information

Supplemental information for this article can be found online at http://dx.doi.org/10.7717/peerj.19356#supplemental-information.

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
