# Peer review of "Simulation of electroporation threshold based on the evolution of transmembrane potential and pore density"

_PeerJ, doi:10.7717/peerj.19356_

## Round 0.1 · original submission · Major Revisions

Please respond to the comments from the 2 expert reviewers

Reviewer 1 ·

Basic reporting

The paper describes a computation for transmembrane potential including membrane conductivity modification due to pores.
It has to be extensively improved and revised to clarify some sentences.
The paper has to be improved following comments in the text. Some statement have to be verifid in the existing litterature.
English revision required.
Please, improve figure. character size and clarity

Experimental design

The simulation design has to be better described. It lack of some particulars.

Validity of the findings

Results has to be validated. This part is not clear. Please, check mesh size in critical regions.

Additional comments

Please, consider the comment in the attached file

Annotated reviews are not available for download in order to protect the identity of reviewers who chose to remain anonymous.

Reviewer 2 ·

Basic reporting

1. Why do the authors use the term “microporous”? Pores are not microscale, so it seems odd to use that terminology. Pore density is more common (I have also seen “nanopores” since they tend to be nanoscale, but that seems superfluous and is commonly used for the small pore generated by nanosecond pulses). It would be good to include some statement on average pore radius or such for completeness as pore size can also be an important metric for understanding pulsed electric field effects.
2. Please state the transmembrane and number density thresholds for electroporation in the Abstract.
3. Given how many studies have been conducted on electroporation over the past three decades, the number of references is very low. Some points:
a. Include references from the modeling work performed by Ravi Joshi and Jim Weaver
b. More recently, Ravi Joshi’s group has recently used COMSOL to explore electroporation and permeabilization of mitochondrial membranes and the endoplasmic reticulum. Specifically:
i. W. Milestone, C. Baker, A. L. Garner, and R. P. Joshi, “Electroporation from mitochondria to cell clusters: model development towards analysis of electrically driven bioeffects over a large spatial range,” J. Appl. Phys. 133, 244701 (2023).
ii. C. Baker, A. Willis, W. Milestone, M. Baker, A. L. Garner, and R. P. Joshi, “Numerical assessments of geometry, proximity and multi-electrode effects on electroporation in mitochondria and the endoplasmic reticulum to nanosecond electric pulses,” Sci. Rep. 14, 23854 (2024).
c. Refs. [2]-[5] are old review papers – there are more recent ones in each topic.
4. Concerning the papers in Comment 3.b.i and 3.b.ii, the authors need to explicitly state how their work differs. I think there are a few key points of distinction, but these need to be spelled out.
5. Ref. [11] – use the article number (page numbers do not identify this journal since it is identified by the article number). Ref. [19] needs to include all page numbers.
6. Fig. 3 – Please make the text larger. Also, since the authors included numbers on the axes, units should also be included.
7. The definition of arc length should be clearly denoted in Fig. 3. Where is 0? I presume it is from the direction of the pulse, but where is that? I am used to this being reported as some angle theta, but that does not really work for nonuniform structures like the ER, so arc length is fine; however, it needs to be defined where 0 is.
8. Fig. 5 – the figures must be able to stand alone. Are the times referring to time during a pulse? Fig. 5d and 5f – I only see one curve. Is there no change with time? This should be explicitly stated.
9. Fig. 6 – The vertical axis of panel (b) needs to be labeled and have units. Since the authors have the pore density along the arc length, have the authors considered including the transmembrane potential along the arc length? That could be easier to interpret than panel (a) alone.
10. Fig. 8 – Not sure what happened, but the resolution of the figures when copied into the document is poor. Also, it is difficult to see some of the colors (Particularly yellow). Also – pick one pulse waveform and perform the comparison across all behavior. It is odd to change (for instance, look at TMP at one electric field and pore density at another electric field). Finally, the x-axes labels for panels (c) and (d) are missing.
11. Table 1: The nuclear envelope thickness is not typically the same as the cell membrane thickness since it is a double membrane. See the work by Yuri Feldman for typical values. The conductivity of the nucleoplasm and cytoplasm are NOT the same. See Yuri Feldman’s papers for typical values. Since the authors chose this to compare to a previous study, it is okay, but they should include statements about more common values that come from measurement (Feldman’s papers are the typical ones used). Please include the reference for the values used in the Table in the Table title.

Experimental design

While many models have performed to assess the mechanism of electroporation, one key challenge is applying them to more realistic cell geometries, particularly intracellular structures. While some studies have begun to couple the asymptotic Smoluchowski equation with COMSOL to assess these structures (please see comments, add the references, and emphasize the differences here), the extension to more accurate cellular geometries and benchmarking to experiments remains incomplete. This manuscript takes an important first step in this process by mapping an actual cell into COMSOL and examining the pore density and transmembrane potential at the cell membrane, nuclear envelope, and endoplasmic reticulum.

I am supportive of publication after addressing the comments in the other sections of this review. Most of them involve clarification of some figures, describing potential variation in selected parameters, and adding some more references to strengthen the manuscript.

Validity of the findings

The results are reasonable, but a few points could be added to strengthen them.
1. One general point that I have noted in my recent submissions is an increasing push toward comparing to experimental results, which can be challenging for these models. While I do not think that is readily done here, I suggest the authors consider adding a statement in the Conclusion about future experiments with realistic cells to explore membrane permeabilization for the membrane and intracellular structures.
2. Concerning electroporation conditions, Wyss et al. [https://doi.org/10.1109/TBME.2024.3471413] recently related the asymptotic Smoluchowski equation to experimental results to conceptually relate pore density and fractional pore area to experimental conditions. It may be worth looking at this assessment for the work performed here.
3. The authors need to be careful with statements concerning a single transmembrane potential for electroporation. There is literature that indicates/suggests variation in TMP at electroporation depending on pulse duration. I suggest including a statement and references to this effect for completeness.

Additional comments

1. Line 132: The authors refer to “cosine law.” I think they are referring to the TMP varying with the angle from the applied field, but this is unclear. Pleas explicitly include this equation in the text with an appropriate citation.

---

## Round 0.2 · Minor Revisions

Your article requires a few Minor Revisions. Please address these changes and resubmit.

Reviewer 1 ·

Basic reporting

The paper was improved with respect previous version.
Nevertheless, some remarks to improve more the paper are listed here.
Please, add references to ESOPE protocol.
In figure please add the position of the electrode
About the FEM model please describe accurately the geometry and physic applied and boundary. How the transmembrane poential is computed in FEM? How eq(1)-(6) are computed using FEM?

Experimental design

No more comments

Validity of the findings

The paper is interesting

Reviewer 2 ·

Basic reporting

The authors have essentially addressed my prior comments. I have only a few minor points that came up when reviewing the recent modifications prior to acceptance.

Experimental design

No comment.

Validity of the findings

No comment.

Additional comments

1) The third paragraph of the Introduction has never been true completely, particularly this statement “the basic quantity that determines the threshold of cellular EP is still the field strength [6]” (it may be under specific conditions). Now, if you fix pulse duration then absolutely; however, generally, speaking, this is not true.

The question has always been one of a combination of parameters. Please see this recent paper that examines this in great detail using the asymptotic Smoluchowski equation. One of the major questions through the decades has been the appropriate scaling parameter – people have often used energy. This paper demonstrates that the solution is nuanced and depends on pulse duration. Please update this paragraph since it is misleading.

S. J. Wyss, W. Milestone, R. P. Joshi, and A. L. Garner, “Maps of membrane pore dynamics from picosecond to millisecond pulse durations,” IEEE Trans. Biomed. Eng. 72, 768-776 (2025).

2) Regarding the technique, I suggest the authors just delete “finite element dielectric model” from the sentence describing their approach. It is not a common statement and just adds unnecessary confusion to what the authors are trying to accomplish.

3) Minor point – In one paragraph on p. 5, the authors write “…set to 0 potential” and “…each point on the cell is 0.” In these contexts, “0” should be written as “zero”. Same comment on p. 7, first full paragraph.

4) Fig. 3: Some of the words got cut due to their size in the figure panels. Please adjust as possible.

---

## Round 0.3 · Minor Revisions

It requires a few Minor Revisions as detailed by Reviewer 1

Reviewer 1 ·

Basic reporting

The paper was extensively reviewed.
Just a remark: please, add reference to ESOPE protocol for ECT like e.g. Michel Marty, Gregor Sersa, Jean Rémi Garbay, Julie Gehl, Christopher G. Collins, Marko Snoj, Valérie Billard, Poul F. Geertsen, John O. Larkin, Damijan Miklavcic, Ivan Pavlovic, Snezna M. Paulin-Kosir, Maja Cemazar, Nassim Morsli, Declan M. Soden, Zvonimir Rudolf, Caroline Robert, Gerald C. O’Sullivan, Lluis M. Mir, Electrochemotherapy – An easy, highly effective and safe treatment of cutaneous and subcutaneous metastases: Results of ESOPE (European Standard Operating Procedures of Electrochemotherapy) study, European Journal of Cancer Supplements, Volume 4, Issue 11, 2006, Pages 3-13, https://doi.org/10.1016/j.ejcsup.2006.08.002.

Gehl, J., Sersa, G., Matthiessen, L. W., Muir, T., Soden, D., Occhini, A., … Mir, L. M. (2018). Updated standard operating procedures for electrochemotherapy of cutaneous tumours and skin metastases. Acta Oncologica, 57(7), 874–882. https://doi.org/10.1080/0284186X.2018.1454602

Experimental design

Please, clarify in the method description if the tranmembrane potential is evaluate in a discrete set of points or continously on a line

Validity of the findings

no more comments

Additional comments

no more comments

Reviewer 2 ·

Basic reporting

The authors have addressed the previous comments. This is a good paper, and I recommend publication as is.

Experimental design

The authors have addressed prior comments.

Validity of the findings

The authors have addressed prior comments.

---

## Round 0.4 · accepted · Accept

This manuscript is ready for publication.

Please carefully check citations, figure labels, format, language, and readability once more.

Reviewer 1 ·

Basic reporting

The paper was improved and all comments addressed.

Experimental design

no more comments

Validity of the findings

no more comments

Additional comments

no more comments